# Comparative Research on Tensile Properties of Cement–Emulsified Asphalt–Standard Sand (CAS) Mortar and Cement–Emulsified Asphalt–Rubber Particle (CAR) Mortar

**DOI:** 10.3390/ma13184042

**Published:** 2020-09-11

**Authors:** Chaoyuan Li, Zanqun Liu, Juan Chen, Qiang Yuan

**Affiliations:** 1School of Civil Engineering, Central South University, Tianxin District, Changsha 410083, China; 184811019@csu.edu.cn; 2Anhui Engineering Material Technology Co., Ltd. of Ctce Group, Luyang District, Hefei 230041, China; chenjuan99@crecg.com

**Keywords:** cement, emulsified asphalt, rubber particle, tensile property, interfacial transition zone

## Abstract

The paper compared the tensile strength and elongation at break of cement–emulsified asphalt–standard sand (CAS) mortar and cement–emulsified asphalt–rubber particle (CAR) mortar. The tensile properties of CAS and CAR mortars were investigated. Microscopic analysis was carried out by Environmental Scanning Electron Microscopy and Energy Dispersive Spectrometer. The test results showed that the tensile strength of the CAR mortar at 7 days improved by about 9.09% higher than that of the CAS mortar, and further increased to 17.76% higher at 28 days The values of elongation at break of the CAR mortars at 3 days, 7 days, and 28 days increased by about 70% higher than those of the CAS mortars. Microscopic analysis showed that in the hardened CAS mortar, an obvious bubble accumulation layer with many pores appeared at the interfacial transition zone (ITZ). In the hardened CAR mortar, asphalt wrapped both cement hydration products and rubber particles and formed an integrated structure where a relatively dense and strong ITZ was formed as a result. This paper proves that the CAR system has superior tensile properties and has a promising future in waste rubber disposal.

## 1. Introduction

The rapid development of the automobile industry, along with the development of the global economy, has resulted in a large amount of waste tires every year. Waste rubber is difficult to decompose and dispose of. Landfilling often causes water and soil contaminations, and burning brings about air pollution [1]. How to deal with the problem of waste rubber has become one of the current research hotspots in waste utilization.

Crushing waste rubber tires into particles and powder is a commonly used treatment method to produce a hot-mix asphalt mixture. Under the action of heat and mechanical force, the rubber powder swells by absorbing the oil in the asphalt and changes from a compact structure to a relatively loose flocculent structure, which is more evenly suspended and dispersed in the asphalt solution [2]. It works together with the matrix asphalt to improve the rut resistance, waterproofing, and crack resistance of the mixture and improve the properties of asphalt mixtures at high temperatures and low temperature [3,4,5,6,7].

Another disposal method is applying rubber particles to the pavement concrete, which can improve the toughness and impact resistance of the concrete [8,9,10], absorb sound, and reduce noise [11]. However, the smooth and hydrophobic surface of the rubber particles results in a weak bond with cement hydration products, resulting in a significant reduction in the bending strength of the pavement concrete [12]. Therefore, the surface of the rubber particles must be treated prior to use.

The current surface treatment measures for rubber mainly include precoating cementitious materials; soaking in NaOH, acid, and silane coupling agent solution; ultraviolet radiation; and local oxidation [13]. These methods aim to change the hydrophobicity of the rubber surface and improve the roughness, thereby improving the mechanical properties of the concrete. For example, when the rubber surface is precoated with cement paste or silica fume, the compressive strength at 28 days of rubber concrete can increase by 31% or 25%, respectively [14]. Ultraviolet radiation treatment can increase the flexural strength of rubber concrete by 5–20% [15]. However, these treatment methods usually have complicated procedures and require the use of highly corrosive chemicals, which may cause great environmental pollution and high costs [13]. As a consequence, the practical application of pretreated rubber in pavement construction is restricted.

In the hot-mix asphalt mixture, the rubber particles absorb the oil in the asphalt and swell, forming a whole with the asphalt [2]. The rubber particles are also able to form a whole with emulsified asphalt. It has been proven that emulsified asphalt and cement can work well together. For example, cement–emulsified asphalt (CA) mortar has high toughness but also good vibration resistance and crack resistance [16,17], so it has been widely used in high-speed railway construction in China [18]. Emulsified asphalt not only bonds well with the rubber particles, but also forms a whole with the cement hydration products. In practical cases, the cement, emulsified asphalt, and rubber only need to be mixed together by stirring without heating. In the hardening process, the three can form a cooperative working system where the emulsified asphalt acts as a surface modifier for the rubber particles and also as cementitious material for the pavement concrete. The cement–emulsified asphalt–rubber particle (CAR) system is expected to have the performance characteristics of both asphalt mixture and pavement concrete.

In summary, CAR mortar should have better toughness than CA mortar. In order to verify this assumption, this paper characterized toughness through tensile properties, designed two systems (i.e., cement–emulsified asphalt–rubber particle (CAR) mortar and cement–emulsified asphalt–standard sand (CAS) mortar), and compared the tensile strengths and elongation at break of the two mortar systems. The microstructures and hydration products at the interfacial transition zones (ITZs) of the mortars were analyzed by an environmental scanning electron microscope (ESEM) and energy dispersive spectrometer (EDS).

## 2. Materials and Methods

### 2.1. Materials

Sulfate aluminum cement (SAC; P 42.5, GB20472, China) with an average size of 18 μm and a density of 2.897 g/cm^3^ was used for this research; the main chemical compositions are listed in Table 1. ISO standard sand (GSB08-1337, China) with an average size of 695 μm and a density of 2.547 g/cm^3^ was used. The absorption rate was 1.73%, and the silica content was greater than 96%. Rubber particles with an average size of 713 μm, a density of 0.93 g/cm^3^, and an absorption rate of 0.87% were purchased from Dujiangyan Huayi Rubber Co., Ltd. (Chengdu, China) Figure 1 shows the rubber particles under a 40× optical microscope, and the main chemical compositions are listed in Table 2. Emulsified asphalt (SH/T 0798, China) was purchased from the Anhui Engineering Material Technology co., Ltd. (Luyang, China) of Ctce Group (Hefei, China). Figure 2 shows the particle size distribution of cement, rubber particles, and standard sand.

### 2.2. Methods

#### 2.2.1. Mix Design

As Figure 2 indicates, the standard sand and the rubber particles showed a similar size distribution. The calculated results based on the densely packed model are shown in Figure 3a. The void ratio was smallest when the volume proportion of cement reached 43%. In this case, the mass ratio of cement to rubber particles was 1:0.43, and the ratio of cement to standard sand was 1:1.17. As illustrated in Figure 3b, the Fuller curve of both the experimental group (cement + rubber particles) and the control group (cement + standard sand) was quite close to the ideal curve. 

In the mix design of the CAR mortars, the mass of emulsified asphalt was first determined and kept the same for all mortars. The mass of rubber particles in the CAR mortars was set as 15%, 20%, 25%, and 30% of emulsified asphalt by weight. The mass ratio of cement was then determined in accordance with the mass ratio of cement to rubber particles (1:0.43). In the CAS mortars, the mass of cement was set to be the same as that in the CAR mortars, and thus the mass of standard sand could be determined as the optimum mass ratio of cement to standard sand (1:1.17) based on the Fuller curve. The final mix design is shown in Table 3.

#### 2.2.2. Specimen Preparation

The cement and rubber particles (standard sand) were mixed by slow stirring for 30 s to a uniform solid mixture. The solid mixture was then added to the emulsified asphalt, followed by slow stirring for 30 s, fast stirring for 1 min, and slow stirring for another 30 s to form a uniform slurry. The slurry was immediately cast in a 10 × 50 × 250 mm^3^ mold and cured for 1 day under natural conditions (temperature: 25 ± 1 °C, humidity: 60% ± 10%). After demolding, the mortars were placed at a constant temperature and humidity box (temperature: 20 ± 2 °C, humidity: 60% ± 5%) and cured to the specified age (3 days, 7 days, 28 days) for testing.

#### 2.2.3. Determination of Tensile Strength and Elongation at Break

The test specimens are shown in Figure 4a. Uniaxial tensile testing (Figure 4c) was performed with a universal testing machine (Figure 4b). The fixture spacing was 200 mm. The loading rate was 10 mm/min. When the force peak decreased by 80%, the loading was stopped, and the tensile strength and elongation at break were recorded. The arithmetic mean of 5 samples for each mortar was calculated as the final results.

#### 2.2.4. Microscopic Analysis

(1)Instrument Introduction

An Environmental Scanning Electron Microscope (ESEM; Quanta 200 (FEI Czech Republic S.r.o., Brno, Czech Republic) is shown in Figure 5. The acceleration voltage was 200 V~30 KV, the magnification was 25~10,000, and the secondary electron resolution was 3 nm.

An Energy Dispersive Spectrometer (EDS; Genesis 2000, FEI Czech Republic S.r.o., Brno, Czech Republic)) with spectrometer resolution <131 eV was also used for analysis.

(2)Specimen Processing

Small pieces of the specimens of hardened R-4 and S-4, after being cured for 1 month, were selected for ESEM analysis. The ESEM images were captured by scanning electron microscopy on the fractured surfaces of the samples after gold coating. 

## 3. Results and Discussion

### 3.1. Tensile Strength

The tensile strengths of the test specimens are shown in Table 4 and Figure 6. It can be seen that as the curing age increased, the tensile strengths of all mortars increased. With the increase of the cement content, the tensile strengths of both the CAR and CAS mortars also improved. In Group 1, the CAR and CAS mortars had nearly the same tensile strengths at 7 days; however, for the other Groups, the 7 days tensile strength of the CAR mortars increased by about 9.0% higher than that of the CAS mortars. After 28 days, the average tensile strength of the CAR mortars was 17.76% higher than that of the CAS mortars. The gap in tensile strength at 28 days between the CAR and CAS mortars increased with more cement addition in the mortars. 

### 3.2. Elongation at Break

The elongation at break results are shown in Table 5 and Figure 7. It can be seen that (1) the elongation at break of both the CAR and CAS mortars decreased as the cement content increased; (2) the elongation at break of the CAR mortars was much higher than that of the CAS mortars, especially when the cement content was low; and (3) the 3 days elongation at break of the CAR mortars increased by 70.47% higher than that of the CAS mortars, and the 28 days elongation at break reached 82.12% higher on average.

### 3.3. Microscopic Analysis

From the above results, it can be seen that compared with the CAS mortar system, the CAR mortar system has higher tensile strength and elongation at break, indicating that components in the CAR system are able to coordinate with each other to present better performance.

Obviously, the different microstructures in the CAS and CAR systems are expected to be responsible for their differences in mechanical properties. Therefore, ESEM and EDS were carried out to study their differences in structure and composition at the interface transition zone (ITZ). It was found that the emulsified asphalt softened under high temperature during gold coating (Groups 1, 2, and 3), making the observation of their ITZ impossible. In Group 4, the mortars had the lowest asphalt content, so these mortars (R-4 and S-4) are desirable for microscopic analysis.

Figure 8 and Figure 9 present the ESEM images of S-4 and R-4, respectively. It can be seen that cement hydrates and emulsified asphalt formed a porous matrix where the emulsified asphalt dehydrated and formed a continuous film, thereby bonding the cement hydration products together. The ITZ of the CAS mortar presented a distinct bubble accumulation layer (Figure 8a–d) or crystal accumulation area (Figure 8e). The ITZ had much more but smaller pores than the matrix as shown in Figure 8f. However, in the CAR mortars, the asphalt wrapped not only the cement hydrates but also the rubber particles together, and thus, the ITZ shows a similar structure with the matrix (Figure 9a–c). No obvious bubble accumulation layer was found at the ITZ (Figure 9d,e), and gelatinous material at the ITZ bridged the rubber particles and the matrix closely (Figure 9e,f).

The element compositions of the products at the ITZ were analyzed by EDS. The test results of S-4 and R-4 are shown in Figure 10 and Figure 11.

It can be seen from Figure 10 that the content of Si in the particle was high, indicating that the particle is standard sand. The total contents of Al, S, and Ca at the ITZ reached 12.44 wt.%. As the scanning area increased, the total contents of Al, S, and Ca increased, while the content of C was slightly reduced. This indicates that the ITZ was mainly filled with sulfate aluminum cement (SAC) hydrates and asphalt.

In Figure 11, the main elemental compositions of the particles were C and O, and the C element content was as high as 91.31 wt.%. H is not shown in the EDS [19], so the particle is determined to be rubber. The main element at the ITZ was C, which shows that the main composition is asphalt. The Al, S, Ca elements were also found at the ITZ. The total contents of Al, S, and Ca reached 5.49 wt.%, which was much less than that at the ITZ of the CAS mortar. As the scanning area increased, the total contents of Al, S, and Ca increased a bit, but were still far less than that at the ITZ of the CAS mortar. In contrast, the content of C at the ITZ reached 88.65 wt.%, which was higher than that at the ITZ of the CAS mortar.

In the CAS and CAR systems, when the emulsified asphalt was mixed with cement, the SAC grasped the water in the emulsified asphalt, resulting in the demulsification of the emulsified asphalt. The SAC hydration crystals grew and formed a network structure, which was warped and filled by soft asphalt. The sand has a negatively charged surface, which repels the anionic emulsified asphalt particles [20]. The sand, however, tends to combine with the hydrophilic cement hydration products. Therefore, more cement hydration products were found at the ITZ of the CAS system. In addition, the hydrophobic asphalt left excessive water at the interface, making the ITZ highly porous. As a consequence, the tensile failure of the CAS mortar was more similar to the brittle fracture of ordinary cement mortar, with low tensile strength and small elongation at break. By contrast, rubber is an organic material and shares some similar hydrophobic properties with asphalt. Therefore, not only does asphalt wrap the rubber particles but also cement hydrates together, forming an integrated structure. In this way, the CAR system has a relatively dense and strong ITZ, thus presenting greater tensile strength and elongation at break than the CAS system. 

## 4. Conclusions

In this paper, a mechanical performance test and microanalysis were carried out to study the tensile properties of the CAR mortar, and the following conclusions can be drawn:The tensile strength of the CAR mortar is 9.09% higher than that of the CAS mortar at 7 days and further grows to 17.76% higher at 28 days. The elongation at break can be increased by more than 70%. The CAR system has excellent tensile properties.In the CAR mortar, asphalt wraps both cement hydration products and rubber particles, forming a dense and strong ITZ, which improves the tensile strength and elongation at break of the CAR mortar.The ternary system of cement, emulsified asphalt, and rubber particles has the potential to provide a reasonable disposal method for waste rubber.

## Figures and Tables

**Figure 1 materials-13-04042-f001:**
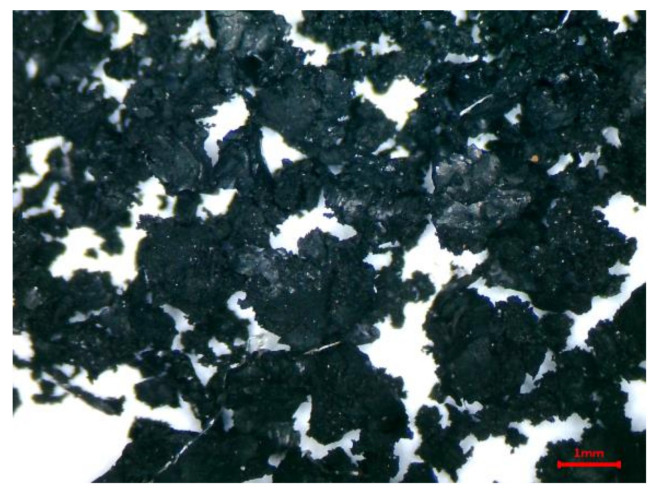
Rubber particles under a 40× optical microscope.

**Figure 2 materials-13-04042-f002:**
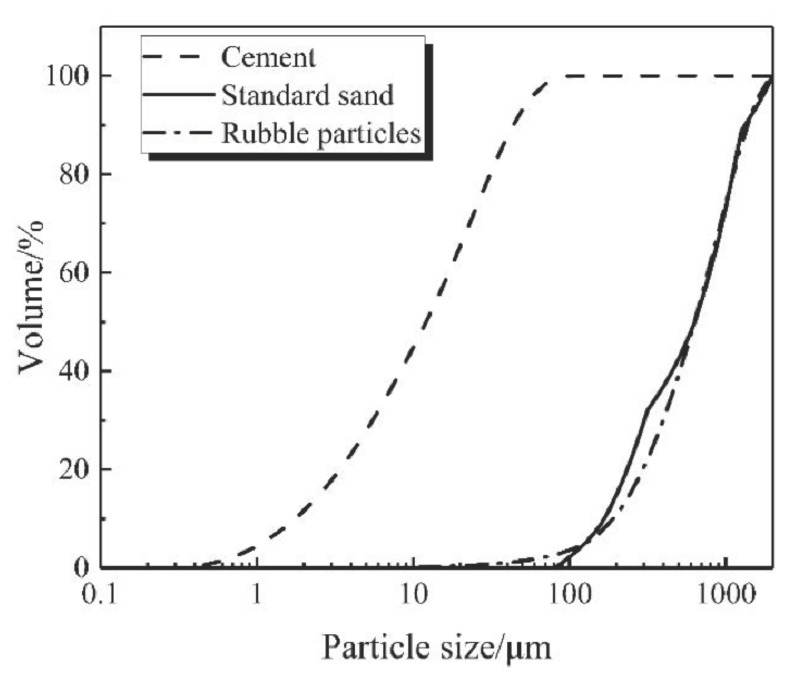
Particle size distribution of materials.

**Figure 3 materials-13-04042-f003:**
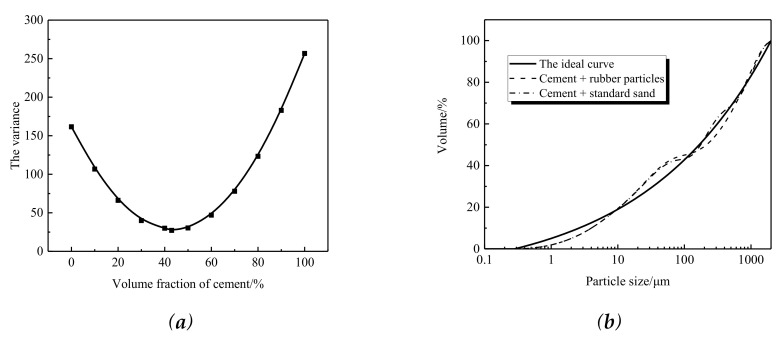
Fuller gradation curve: (**a**) calculation of variance; (**b**) Fuller curve.

**Figure 4 materials-13-04042-f004:**
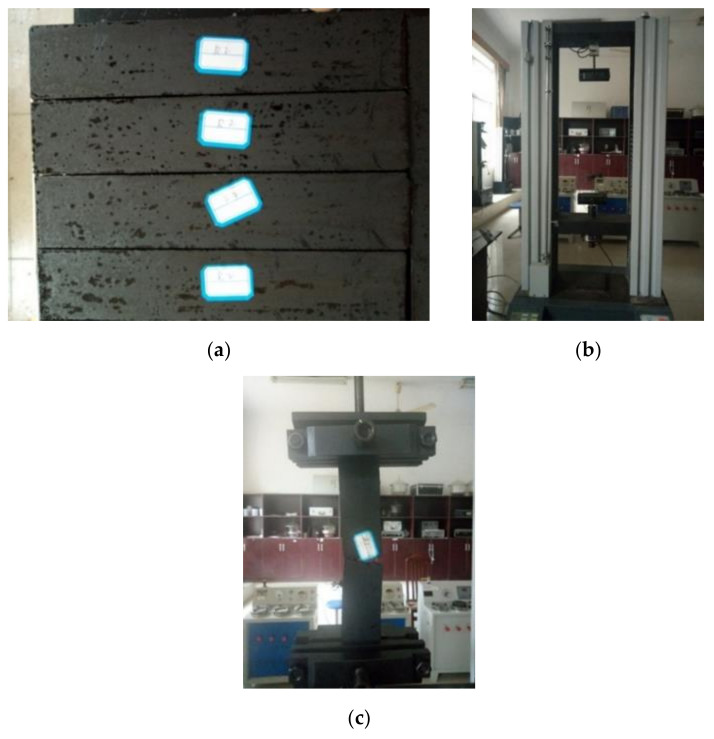
Images of the uniaxial tensile test: (**a**) test specimen; (**b**) testing machine; (**c**) testing.

**Figure 5 materials-13-04042-f005:**
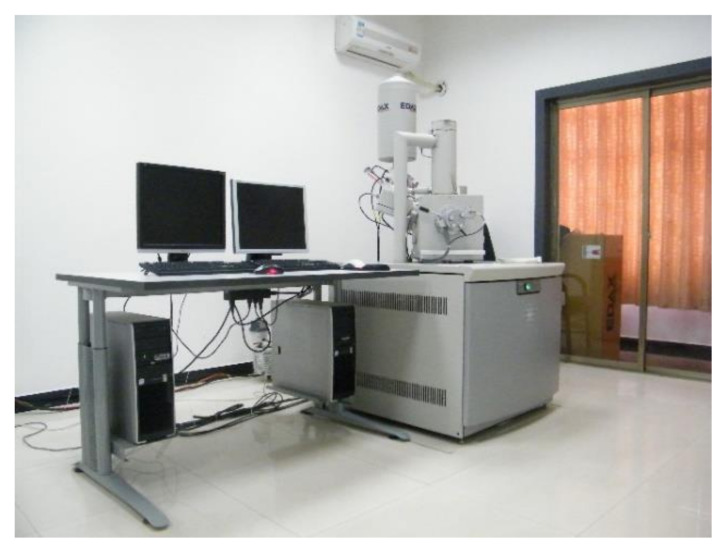
Environmental scanning electron microscope (Quanta 200 (FEI Czech Republic S.r.o., Brno, Czech Republic)).

**Figure 6 materials-13-04042-f006:**
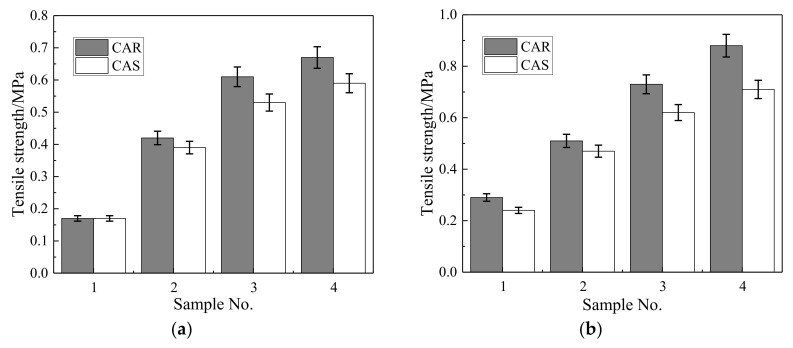
Comparison of tensile strengths between the CAS and CAR mortars at different ages: (**a**) 7 days curing age; (**b**) 28 days curing age.

**Figure 7 materials-13-04042-f007:**
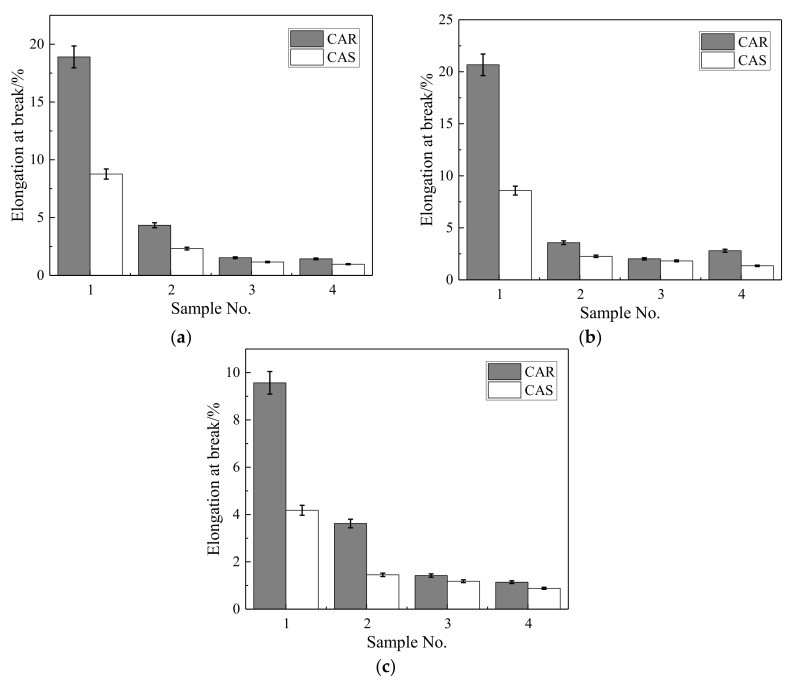
Comparison of elongation at break between rubber particles and standard sand at different ages: (**a**) 3 days curing age; (**b**) 7 days curing age; (**c**) 28 days curing age.

**Figure 8 materials-13-04042-f008:**
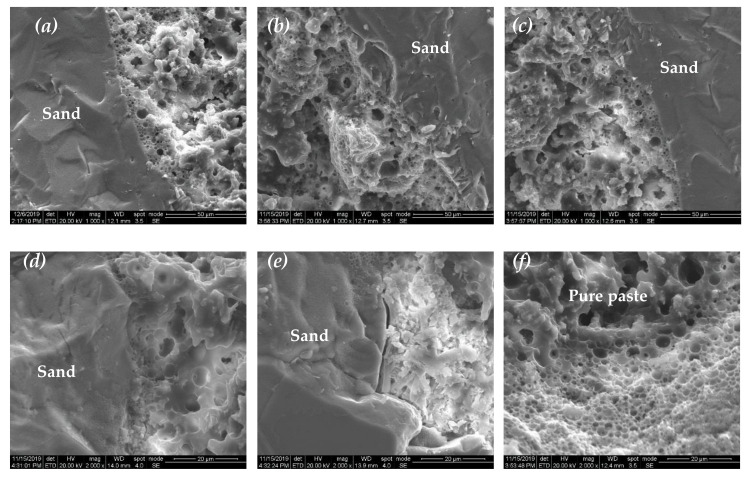
(**a**–**f**): ESEM images of the cement–emulsified asphalt–standard sand (CAS) mortar (S-4).

**Figure 9 materials-13-04042-f009:**
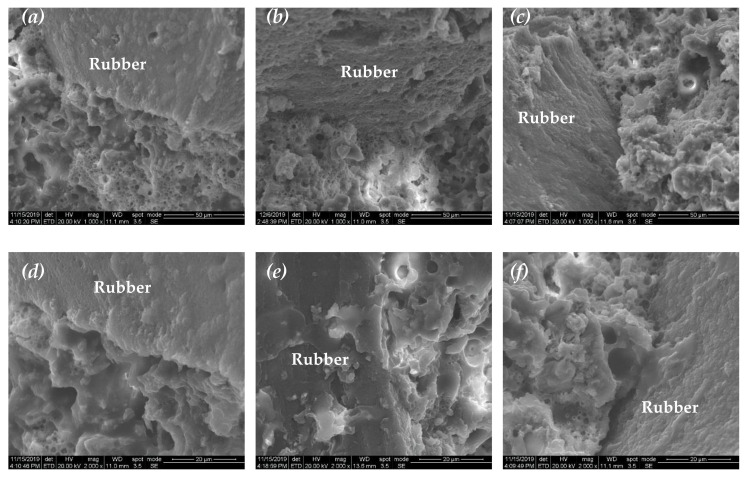
(**a**–**f**): ESEM images of the cement–emulsified asphalt–rubber particle (CAR) mortar (R-4).

**Figure 10 materials-13-04042-f010:**
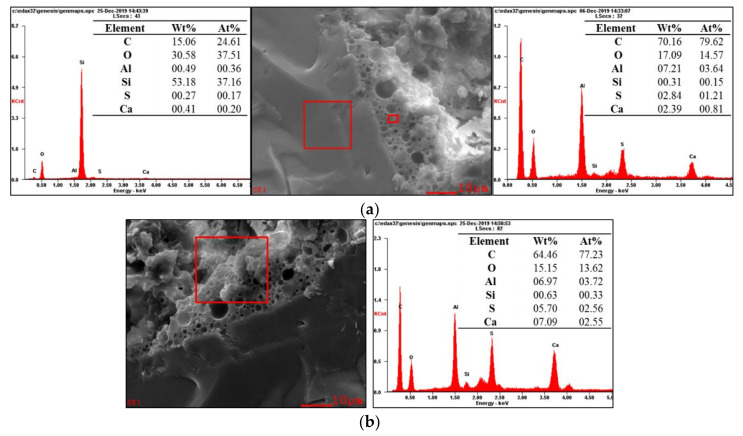
Analysis of the element compositions at the interfacial transition zone (ITZ) of the CAS mortar (S-4): (**a**) sand and paste; (**b**) paste in a larger scanning area.

**Figure 11 materials-13-04042-f011:**
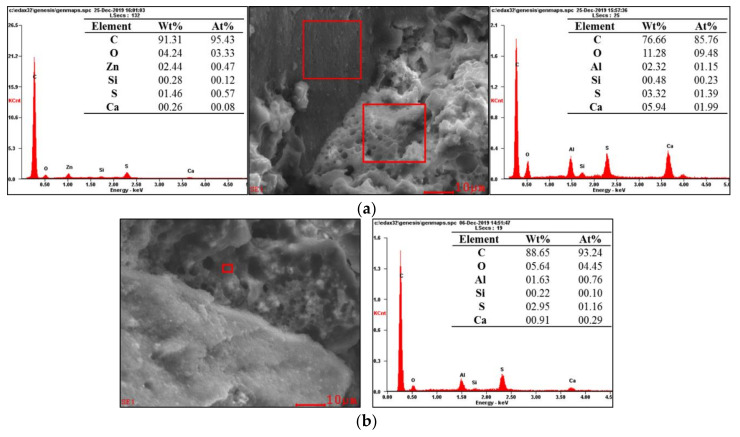
Analysis of the element compositions at the ITZ of the CAR mortar (R-4): (**a**) rubber and paste; (**b**) paste in a smaller scanning area.

**Table 1 materials-13-04042-t001:** Main chemical compositions of sulfate aluminum cement.

Compositions	CaO	SiO_2_	Al_2_O_3_	Fe_2_O_3_	SO_3_	MgO
Mass fraction/%	62.21	19.12	5.39	3.79	3.06	0.86

**Table 2 materials-13-04042-t002:** Main chemical compositions of rubber particles.

Compositions	Ash	Acetone Extract	Carbon Black	Rubber Hydrocarbon
Mass fraction/%	5.38	10.16	28.78	54.45

**Table 3 materials-13-04042-t003:** Mix design of the CAS and CAR mortars.

Groups	Sample No.	Emulsified Asphalt	Rubber Particles	Standard Sand	Cement
1	R-1	1	0.15		0.35
	S-1	1		0.41	0.35
2	R-2	1	0.2		0.47
	S-2	1		0.55	0.47
3	R-3	1	0.25		0.58
	S-3	1		0.69	0.58
4	R-4	1	0.3		0.7
	S-4	1		0.82	0.7

**Table 4 materials-13-04042-t004:** Comparison of tensile strengths between the CAS and CAR mortars.

Groups	Sample No.	Tensile Strength/MPa		
		3 Days	7 Days	28 Days
1	R-1	0.13	0.17	0.29
	S-1	0.14	0.17	0.24
2	R-2	0.16	0.42	0.51
	S-2	0.19	0.39	0.47
3	R-3	0.49	0.61	0.73
	S-3	0.47	0.53	0.62
4	R-4	0.57	0.67	0.88
	S-4	0.52	0.59	0.71

**Table 5 materials-13-04042-t005:** Comparison of the elongation at break between the CAS and CAR mortars.

Groups	Sample No.	Elongation at Break/%		
		3 Days	7 Days	28 Days
1	R-1	18.9	20.67	9.57
	S-1	8.77	8.58	4.18
2	R-2	4.34	3.57	3.62
	S-2	2.32	2.26	1.45
3	R-3	1.53	2.02	1.42
	S-3	1.16	1.82	1.18
4	R-4	1.43	2.80	1.14
	S-4	0.97	1.35	0.88

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
