# Peer review of "Comparative Research on Tensile Properties of Cement–Emulsified Asphalt–Standard Sand (CAS) Mortar and Cement–Emulsified Asphalt–Rubber Particle (CAR) Mortar"

_materials, 2020, doi:10.3390/ma13184042_

Round 1
Reviewer 1 Report
The manuscript deals with an issue of cement-emulsified asphalt-rubber particle mortar. I evaluate the topic as interesting and beneficial. The conclusions of the study are clear, interesting and useful.
I see the only problem that the manuscript is too short - an article in a scientific journal is usually longer and contains more results (comprehensive study of the problem). Therefore, I recommend adding more results to the manuscript - there are many other properties of the investigated material that can be verified.
After expanding the article, I recommend manuscript for publication.
Author Response
Thank you for your comments. All of your comments are very important. They are of important guiding significance for my thesis writing and scientific research.
Point 1: I see the only problem that the manuscript is too short - an article in a scientific journal is usually longer and contains more results (comprehensive study of the problem). Therefore, I recommend adding more results to the manuscript - there are many other properties of the investigated material that can be verified.
Response 1: Thank you for your advice.
The original intention of this study was to demonstrate that CAR mortar has better mechanical properties than CAS mortar. In the initial stage, we also tried to study the flexural strength and compressive strength of the two mortars. But as a result of emulsified asphalt in this experiment design content is higher, the specimen is a kind of elastic material, compressive strength and flexural strength can't use the general methods for testing. So we switched to the tensile performance to evaluate the performance of the two mortars, and the microscopic morphology analysis of the specimens finally proved our conjecture.
Thank you for your suggestion, it is very important. Because of your suggestion, I found the shortcomings in my current work. I will follow your suggestions to improve the level of scientific research and achieve more results in future work.
Reviewer 2 Report
In article compared and analyzed the properties the tensile strength and elongation at break of cement-emulsified asphalt-standard sand (CAS) mortar and cement-emulsified asphalt-rubber particle (CAR) mortar. The conclusions were correctly formulated.
Page 3, Figure 2, I suggest changing the semicolon (;) to a comma (,) before b
Page 4, Figure 3, I suggest changing the semicolon (;) to a comma (,)
Page 4, I suggest you provide the serial number of the device (ESEM)
Page 4, Table 3, I suggest moving the table caption to page 5
Page 5, Figure 4 I suggest changing the semicolon (;) to a comma (,)
Page 6, Figure 5 I suggest changing the semicolon (;) to a comma (,)
The caption for figure 6 may not be on a page other than the figure. suggests changing that
Page 7, there is no gap between figure 8 and the text
Author Response
I am very grateful to your comments for the manuscript. According with your advice, we amended the relevant part in manuscript. Some of your questions were answered below.
Point 1: Page 3, Figure 2, I suggest changing the semicolon (;) to a comma (,) before b;
Page 4, Figure 3, I suggest changing the semicolon (;) to a comma (,);
Page 5, Figure 4 I suggest changing the semicolon (;) to a comma (,);
Page 6, Figure 5 I suggest changing the semicolon (;) to a comma (,).
Response 1: All the semicolons (;) above have been changed to commas (,).
Point 2: Page 4, I suggest you provide the serial number of the device (ESEM).
Response 2: Sorry, I cannot find the serial number of the device (ESEM). As compensation, I added a picture of ESEM to the manuscript.
Point 3: Page 4, Table 3, I suggest moving the table caption to page 5.
Response 3: The table caption has been moved.
Point 4: The caption for figure 6 may not be on a page other than the figure. suggests changing that.
Response 4: The caption for figure 6 has been moved to the same page with the figure.
Point 5: Page 7, there is no gap between figure 8 and the text.
Response 5: A gap has been added between figure 8 and the text.
Reviewer 3 Report
In this paper, authors compare the tensile strength and elongation at break of cement-emulsified asphalt-standard sand (CAS) mortar and cement-emulsified asphalt-rubber particle (CAR) mortar. The data is good quality and the experiments straight forward and well designed but I have several comments:
Please fill in the Materials section with an information about the rubber particles. How are these particles obtained? What is the shape of these particles? A microscopy image (optical or electron) is required.
Figure 4 - figure quality can be improved. For an easier understanding of the data the representation should be made using the sample NO. (R-1, S-1) not the groups. Also, the error bar is missing.
Figure 5 - the same recommendations as in the case of figure 4
Figure 6 and figure 7 - What specimens are presented in the pictures? Specify in the legend what a), b).... means.
Author Response
Thank you for your comments. All of your comments are very important. They are of important guiding significance for my thesis writing and scientific research. We have revised our paper according to your comments:
Point 1: Please fill in the Materials section with an information about the rubber particles. How are these particles obtained? What is the shape of these particles? A microscopy image (optical or electron) is required.
Response 1: Table 2 has been added to show main chemical compositions of rubber particles. And the manufacturer of rubber particles has been added. A picture of rubber particles under a 40x optical microscope has been added, from which the shape of these particles can be seen.
Point 2: Figure 4 - figure quality can be improved. For an easier understanding of the data the representation should be made using the sample NO. (R-1, S-1) not the groups. Also, the error bar is missing; Figure 5 - the same recommendations as in the case of figure 4.
Response 2: The quality of the images of Figure 4 and Figure5 have been improved. And the representation has used the sample NO. The error bar has been added.
Point 3: Figure 6 and figure 7 - What specimens are presented in the pictures? Specify in the legend what a), b) .... means.
Response 3: All the pictures shown in Figure 6 are S-4 specimens, and all the pictures shown in Figure 7 are R-4 specimens. The specimen names have been added after the captions of Figure 6 and Figure 7.
a), b) .... are different observation areas of the same specimen, to enhance the credibility of the test results, and have no separate meaning. So we did not specify their meaning in the legend.
Reviewer 4 Report
In this study the tensile properties of cement-emulsified asphalt-standard sand (CAS) mortar was compared with cement-emulsified asphalt-rubber particle (CAR) mortar, then from the micrographs of samples, the formation of phases in the matrix of both mortars was studied and compared. While, EDS analyses was carried out to analyses the chemical composition of phases formed at interface transition zone. This paper is written concisely and briefed well. However the following changes is needed.
- The title of this paper is not appropriate considering the research objective which is to compare the tensile properties of CAS with CAR.
- In the last paragraph of introduction, the author needs to clarify the significance of research. Author need to clarify, why the tensile properties of CAR was compared to CAS.
- In this study, the chemical composition of all the materials especially the rubber and standard sand particles is not given. When it comes to comparison, it is necessary to compare the physical and chemical properties of raw materials before executing the experiments. Secondly, the EDS analyses was conducted on the composites, whose ingredients’ chemical composition is not known.
- Along with the chemical composition, the absorption rate of rubber and standard sand is also not known, which I believe is a necessary data to add in this paper.
- The slurry was cured for 1 day in natural condition, what was the temperature and humidity in that condition?
Author Response
Thank you for your comments. All of your comments are very important. They are of important guiding significance for my thesis writing and scientific research. We have revised our paper according to your comments:
Point 1: The title of this paper is not appropriate considering the research objective which is to compare the tensile properties of CAS with CAR.
Response 1: The title of this paper has been modified to “Comparative Research on Tensile Properties of Cement-Emulsified Asphalt-Standard sand (CAS) Mortar and Cement-Emulsified Asphalt-Rubber Particles (CAR) mortar”.
Point 2: In the last paragraph of introduction, the author needs to clarify the significance of research. Author need to clarify, why the tensile properties of CAR was compared to CAS.
Response 2: The significance of this research has been added in the last paragraph of introduction. We guess that CAR mortar should have better toughness than CAS mortar through introduction. In order to verify this assumption, we characterized toughness through tensile properties, compared the tensile properties of the two mortar systems.
Point 3: In this study, the chemical composition of all the materials especially the rubber and standard sand particles is not given. When it comes to comparison, it is necessary to compare the physical and chemical properties of raw materials before executing the experiments. Secondly, the EDS analyses was conducted on the composites, whose ingredients’ chemical composition is not known.
Response 3: Table 2 has been added to show main chemical compositions of rubber particles. The silica content in standard sand is greater than 96%, which has been added into manuscript. See line 77 of the revised manuscript.
Point 4: Along with the chemical composition, the absorption rate of rubber and standard sand is also not known, which I believe is a necessary data to add in this paper.
Response 4: We have added the absorption rate of rubber particles and standard sand to the paper. The absorption rate of rubber is 0.87%,and the absorption rate of standard sand is 1.73%. See lines 77&78 of the revised manuscript.
Point 5: The slurry was cured for 1 day in natural condition, what was the temperature and humidity in that condition?
Response 5: temperature: 25±1°C, humidity: 60±10%, which has been added into manuscript. See line 116 of the revised manuscript.
Round 2
Reviewer 3 Report
I still recommend rejecting the manuscript as it is in general not up to the level of Polymers. The main issue in this manuscript is that the results and discussion section is very weak.
Author Response
Thank you for your suggestion, our research does have some shortcomings: only the tensile mechanical properties were tested in the macroscopic perspective; the thickness of the interface transition zone was not analyzed in the microscopic perspective, and the microscopic characteristics were not discussed in depth. We will improve the experiment for more in-depth research in the future.
As far as this article is concerned, the overall structure is still clear. The tensile performance test and microscopic analysis can show that CAR mortar has better tensile properties than CAS mortar and explain the reason. To a certain extent, this can provide guidance for the application of rubber particles in cement emulsified asphalt.
Therefore, we sincerely hope that this article can be published in Material. We would appreciate it.
Round 3
Reviewer 3 Report
The paper has been improved and can be published in the journal if the editor agrees!